# Telemedicine management of type 2 diabetes mellitus in obese and overweight young and middle-aged patients during COVID-19 outbreak: A single-center, prospective, randomized control study

Wenwen Yin[1,2©], Yawen Liu[3©], Hao Hu[2], Jin Sun[2], Yuanyuan Liu[2], Zhaoling Wang[2]*

1 China University of Mining and Technology, Xuzhou, Jiangsu Province, China, 2 Department of Endocrinology, The First People's Hospital of Xuzhou, Xuzhou, Jiangsu Province, China, 3 Department of Intensive Care Unit, The First People's Hospital of Xuzhou, Xuzhou, Jiangsu Province, China

© These authors contributed equally to this work.
* wzlnfmk@163.com

## Abstract

### Objective

The coronavirus disease-2019 (COVID-19) pandemic severely affected the disease management of patients with chronic illnesses such as type 2 diabetes mellitus (T2DM). This study aimed to assess the effect of telemedicine management of diabetes in obese and overweight young and middle-aged patients with T2DM during the COVID-19 pandemic.

### Methods

A single-center randomized control study was conducted in 120 obese or overweight (body mass index [BMI] $\geq$ 24 kg/m$^2$) young and middle-aged patients (aged 18–55 years) with T2DM. Patients were randomly assigned to the intervention (telemedicine) or control (conventional outpatient clinic appointment) group. After baseline assessment, they were home isolated for 21 days, received diet and exercise guidance, underwent glucose monitoring, and followed up for 6 months. Glucose monitoring and Self-Rating Depression Scale (SDS) scores were evaluated at 22 days and at the end of 3 and 6 months.

### Results

Ninety-nine patients completed the 6-month follow-up (intervention group: n = 52; control group: n = 47). On day 22, the fasting blood glucose (FBG) level of the intervention group was lower than that of the control group (p < 0.05), and the control group's SDS increased significantly compared with the baseline value (p < 0.05). At the end of 3 months, glycated hemoglobin (HbA1c) and FBG levels in the intervention group decreased significantly compared with those in the control group (p < 0.01). At the end of 6 months, the intervention group showed a significant decrease in postprandial blood glucose, triglyceride, and low-density lipoprotein cholesterol levels as well as waist-to-hip ratio compared with the control

**Data Availability Statement:** All data are available from the https://clinicaltrials.gov/ (NCT number: 04723550).

**Funding:** This study was supported by grant no. 2020QN80 from the Fundamental Research Funds for the Central Universities. The funders had no role in study design, data collection and analysis, decision to publish, or preparation of the manuscript.

**Competing interests:** The authors have declared that no competing interests exist.

group ($p < 0.05$); moreover, the intervention group showed lower SDS scores than the baseline value ($p < 0.05$). Further, the intervention group showed a significant reduction in BMI compared with the control group at the end of 3 and 6 months ($p < 0.01$).

## Conclusion

Telemedicine is a beneficial strategy for achieving remotely supervised blood glucose regulation, weight loss, and depression relief in patients with T2DM.

## Trial registration

ClinicalTrials.gov Identifier: NCT04723550.

## Introduction

The rapid outbreak of coronavirus disease-2019 (COVID-19) adversely affected the daily life of people worldwide. Due to the spread of the disease at a pandemic level, hospitals in China implemented strict control measures, including limiting outpatient visits and inpatient admissions as well as reducing operations to avoid cross-contamination caused by personnel movement [1]. However, these epidemic prevention and control measures restricted the patients' access to outpatient follow-up, blood glucose monitoring, and drug supply. Patients with diabetes mellitus were greatly affected during this period in terms of their self-efficacy and management ability [2]. Khare and Jindal followed up with 143 subjects who stayed at home for 3 months due to the nationwide lockdown and found that 56 (39.16%) had significantly elevated blood glucose levels and required additional medication [3]. Verma et al. revealed that the mean glycated hemoglobin (HbA1c) level of 52 patients during COVID-19 isolation (10% ± 1.5%) was significantly higher than their pre-pandemic mean value (8.8% + 1.3%) [4]. In addition, patients with diabetes are more likely to contract COVID-19 and exhibit a poor prognosis [5–7]. Therefore, it is necessary to use the existing limited medical resources to ensure appropriate follow-up and management of patients with diabetes during the COVID-19 pandemic.

Telemedicine refers to remote diagnosis, treatment, and consultation of patients using remote communication, digital holography, modern electronic technology, and computer multimedia [8]. It allows the patient to avail full benefits of medical technology and equipment available at higher medical centers as well as receive disease-related counseling and education when social interaction or direct contact with the healthcare providers is not possible [9]. Arabi et al. has demonstrated that telemedicine had a positive influence on blood glucose control in patients with diabetes during the COVID-19 pandemic [10]. Notably, sudden lifestyle changes that occurred during the COVID-19 lockdown in terms of restriction of outdoor movement and social interaction had a great impact on the young and middle-aged population, which may have led to physical and mental diseases, such as obesity, depression, and anxiety [11, 12]. Therefore, this study aimed to assess the effect of telemedicine on glycemic control and diabetes-related anxiety symptoms in obese and overweight young and middle-aged patients with type 2 diabetes mellitus (T2DM) during the COVID-19 pandemic.

## Methods

### Ethical considerations

This single-center, parallel-group randomized control study was conducted as per the "The Code of Ethics of the World Medical Association (Declaration of Helsinki)" and was approved

by the Ethics Committee of Affiliated Hospital of China University of Mining and Technology (xyy11[2020]40). The patients consented to the procedure, and written informed consent was obtained from them before enrollment. The trial was registered with ClinicalTrials (NCT04723550).

## Study patients and recruitment

In January 2021, the government imposed a strict lockdown due to the COVID-19 outbreak in certain areas of China. For patients with T2DM who had visited our outpatient endocrine clinic before the pandemic, we conducted a telephone follow-up interview. These patients were considered candidates for this randomized controlled trial. Patients fulfilling the following criteria were included in this study: physician diagnosis of T2DM for >6 months [13]; HbA1c level of 7.0%–10.0%; quarantined for 21 days due to COVID-19; aged 18–55 years; body mass index (BMI) $\geq$ 24 kg/m$^2$; and can use smartphone and internet. The exclusion criteria were as follows: insulin pump users; patients with a history of symptomatic cardiovascular disease (myocardial infarction, angina pectoris, surgical or endovascular intervention, stroke older than 6 months, or symptomatic lower limb arteritis); pregnant or lactating women or those who became pregnant during the study; patients who underwent obesity surgery for >1 year; those diagnosed with COVID-19 infection; those with other comorbidities, such as chronic heart disease, cerebrovascular disease, HIV/AIDS, cancer, emphysema, chronic liver or kidney disease, which may affect the patients' ability to follow the tailored advice.

## Patient randomization and data collection procedure

All patients were randomly assigned to the intervention or control group using a random number sequence generated using the SPSS software (version 17.0; IBM Corp., Armonk, NY) in batches of six patients. After enrollment, all patients underwent an initial physical examination and blood sample collection, followed by a mandatory home quarantine for 21 days. During the isolation period, the control group was followed up through telephone once a week.

1. Glucose data management: Postprandial blood glucose (PBG) and fasting blood glucose (FBG) levels were monitored using a glucometer. Patients in the intervention group were provided training for independently using the hospital's telemedicine app. The glucometer was connected to the patient's mobile phone via Bluetooth. The glucometer data were then automatically transferred to the hospital telemedicine app. The patients were followed up four times a week (at least once during the weekends) in the first 3 months and twice a week (at least once during the weekends) in the next 3 months. Doctors reminded patients in the intervention group to monitor their blood glucose levels and provided medical advice through the telemedicine system.
Patients in the control group were followed up through conventional outpatient clinic appointments every 2 weeks, and telephone follow-up was used during the isolation period.

2. Diet guidance and exercise advice: For the intervention group, the dietitian advised the patients on energy intake and food exchange methods. They were provided custom-tailored dietary recommendations and were asked to consume the required calories and upload their daily dietary intake on the telemedicine app. Additionally, the app recorded the patients' daily steps and automatically transferred them to the medical server. Further, exercise guidance was provided to each patient in the intervention group.
The control group received traditional health education, which included diet, exercise, and medication guidance, during clinic visits.

All patients were provided outpatient care and were followed up by the same medical team at 22 days, 3 months, and 6 months after enrollment.

## Outcome measurement

To assess the patients' progress, the following data were reviewed four times (at baseline, 21 days after enrollment, and 3 and 6 months after enrollment): treatment, physical examination, laboratory investigations, and Self-Rating Depression Scale (SDS) scores. Additionally, the number of hypoglycemic episodes and the number of patients who dropped out of the study due to any adverse event were calculated. The SDS is a self-reported 20-item questionnaire assessing depressive state—each item is ranked on a 4-point response scale (1 indicates "a little of the time" and 4 denotes "most of the time") [14]. The total score ranges from 20 to 80, and the reference value for normal SDS in adults is <50.

## Sample size

A previous observational study involving 10 patients reported a decrease in the HbA1c levels by 2.41% (standard deviation [SD] = 1.38%) in the outpatient group and 1.67% (SD = 1.15%) in the telemedicine group after 6 months of follow-up. Based on these results, we considered a bilateral $\alpha$ = 0.05 to achieve 80% test power and an equal ratio of patient allocation to the control and intervention groups (1:1) and computed a target sample size of 48 patients in each group using PASS 15.0 (NCSS LLC). Further, accounting for a 20% loss of sample to follow-up or refusal to follow-up, a final sample size of 60 patients in each group was considered.

## Statistical analysis

All data were analyzed using the SPSS Statistics software (version 17.0; IBM Corp., Armonk, NY). Normally distributed variables are expressed as means and SD, and non-normally distributed variables are expressed as median and interquartile range (IQR). Between-group differences for normally distributed variables were assessed using an independent samples t-test, whereas those for non-normally distributed variables were assessed using the Mann–Whitney U test. For within-group comparisons, normally distributed variables were tested using a paired t-test, whereas non-normally distributed variables were tested using the Wilcoxon signed rank test. The figures were created using GraphPad software (GraphPad Prism version 8.4.2).

The primary outcome, HbA1c level, was evaluated at the four aforementioned time points using two-way repeated-measures analysis of variance (ANOVA). The group effect, time effect, and the effect of the interaction between group and time (group × time) were compared between and within the groups. Bonferroni correction was used for post-hoc comparison. A p-value of <0.05 was considered statistically significant for all analyses.

# Results

Of the 201 potentially eligible patients, 120 expressed interest and were screened for suitability for inclusion in this study. Of these, 6 and 11 patients in the intervention and control groups voluntarily withdrew from the study, respectively, and two in each group were lost to follow-up. A total of 47 and 52 patients in the control and intervention groups completed the 6-month follow-up, respectively (Fig 1). According to the COVID-19 diagnostic criteria, no confirmed cases of COVID-19 were found in this study [15, 16]. The baseline characteristics of the patients are summarized in Table 1. There were no significant between-group differences in terms of age, physical findings, or biochemical indices.

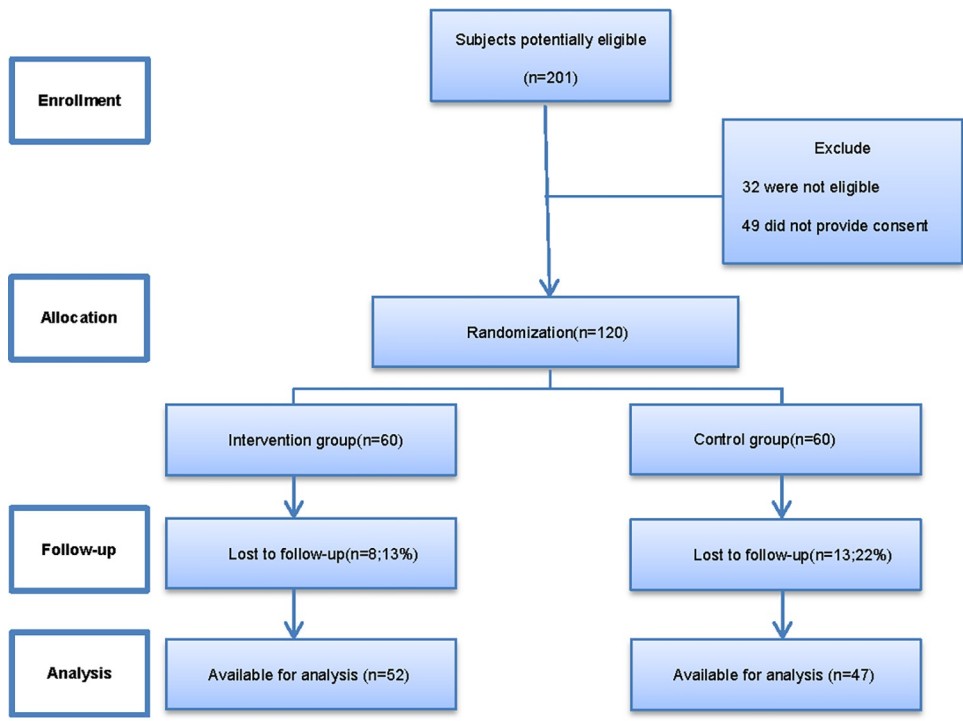

**Fig 1. Patient flow diagram.**

Table 2 presents the summary statistics for outcome variables at the four follow-up time points. Fig 2 shows a significant decrease in the HbA1c, FBG, PBG, triglyceride (TG), low-density lipoprotein cholesterol (LDL-C) levels as well as waist-to-hip ratio (WHR) and BMI in the two groups at the end of 6 months, revealing a statistically significant difference from the baseline values. The median FBG level in the intervention group was lower than the baseline value (median [IQR] at 21 days: 6.52 [5.53–8.17] vs. baseline: 8.45 [7.69–9.35]; p < 0.01) and lower than that in the control group (intervention: 6.52 [5.53–8.17] vs. control: 8.86 [7.77–9.77]; p < 0.01) at the end of 21-day isolation (Table 2, Fig 2B). Compared with the baseline, a statistically significant decrease was observed in the HbA1c, FBG, PBG, TG, and LDL-C levels in both groups and BMI in the intervention group at the end of 3 months (Table 2, Fig 2). Furthermore, the improvement in the HbA1c and FBG levels in the intervention group was better than that in the control group, and the difference was statistically significant (Table 2, Fig 2A and 2B). At the end of the study (6-month follow-up), the extent of decrease in PBG, TG, and LDL-C levels as well as WHR in the intervention group was larger and more significant than that in the control group (Table 2, Fig 2).

The extent of reduction in BMI in the intervention group was significantly greater than in the control group at the 3- and 6-month follow-ups (mean [SD] at 3-month follow-up: 27.10 [2.61] vs. baseline: 29.25 [2.93], p < 0.01; intervention group: 27.10 [2.61] vs. control group: 28.82 [2.57], p < 0.01) (6-month follow-up: 25.49 [2.35] vs. baseline: 29.25 [2.93], p < 0.01; intervention group: 25.49 [2.35] vs. control group: 27.36 [1.90], p < 0.01) (Table 2, Fig 2H). There was no statistically significant difference between the groups regarding blood pressure, total cholesterol, high-density lipoprotein cholesterol, blood urea nitrogen, and creatinine levels as well as estimated glomerular filtration rate at the 6-month follow-up.

Compared with the baseline, the mean SDS score in the control group increased after 21 days of isolation (44.02 [9.71] vs. 40.17 [11.60]; p < 0.05), whereas it decreased in the

**Table 1. Baseline characteristics of the two groups.**

| Characteristic | Control (n = 47) | Intervention (n = 52) | P-value |
|---|---|---|---|
| Age (years) | 47.00 (42.00–51.00) | 47.50 (43.00–51.00) | 0.64 |
| Gender, male, n (%) | 20 (38) | 20 (43) | 0.68 |
| Diabetes mellitus, duration in years | 3.00 (2.00–5.00) | 4.00 (2.00–6.00) | 0.08 |
| FBG (mmol/L) | 8.95 (8.31–9.46) | 8.45 (7.69–9.35) | 0.10 |
| PBG (mmol/L) | 12.76 (11.63–14.22) | 12.73 (11.95–13.79) | 0.64 |
| HbA1c (%) | 8.50 (0.80) | 8.56 (0.88) | 0.75 |
| Blood pressure systolic (mm Hg) | 138.10 (125.20–144.85) | 135.75 (127.43–147.90) | 0.90 |
| Blood pressure diastolic (mm Hg) | 87.90 (80.45–92.10) | 87.10 (79.80–92.10) | 0.61 |
| BMI (kg/m$^2$) | 29.05 (3.31) | 29.25 (2.93) | 0.76 |
| Waist-to-hip ratio | 0.94 (0.04) | 0.96 (0.04) | 0.11 |
| TC (mmol/L) | 5.03 (4.59–5.34) | 4.78 (4.60–5.51) | 0.85 |
| TG (mmol/L) | 1.99 (1.69–2.42) | 1.98 (1.73–2.42) | 0.85 |
| LDL-C (mmol/L) | 3.60 (0.28) | 3.70 (0.28) | 0.08 |
| HDL-C (mmol/L) | 1.30 (1.15–1.41) | 1.20 (1.08–1.34) | 0.06 |
| BUN (mmol/L) | 6.00 (5.50–6.40) | 6.00 (5.40–6.40) | 0.94 |
| Cr (mmol/L) | 64.10 (58.95–69.20) | 62.65 (59.18–69.33) | 0.82 |
| e-GFR(ml/min) | 90.07 (9.15) | 91.53 (9.11) | 0.43 |
| SDS | 40.17 (11.60) | 40.63 (11.10) | 0.81 |

Data are presented as the means (standard deviation) for the normally distributed variables, the median (interquartile range) for the non-normally distributed variables or the number of participants (%). FBG: fasting blood glucose, PBG: postprandial blood glucose, HbA1c: glycated hemoglobin, BMI: Body Mass Index, TC: total cholesterol, TG: triglyceride, LDL-C: low-density lipoprotein cholesterol, HDL-C: high-density lipoprotein cholesterol, BUN: blood urea nitrogen, Cr: creatinine, e-GFR: estimated glomerular filtration rate, SDS: Self-Rating Depression Scale

intervention group at the 6-month follow-up (37.08 [9.16] vs. 40.63 [11.10], p < 0.05). Moreover, the score in the intervention group was lower than that in the control group. These differences showed a decreasing trend but were not statistically significant (Table 2, Fig 2G).

## Trends noted in HbA1c levels during follow-up (Table 3)

The two-way repeated-measures ANOVA revealed that the main effect of the group was not significant (F = 2.558, p = 0.113, partial $\eta^2$ = 0.026), whereas time (F = 86.089, p < 0.001, partial $\eta^2$ = 0.470) and group × time (F = 3.498, p = 0.019, partial $\eta^2$ = 0.035) showed significant effects. The separate group effect was not significant at baseline (F = 0.106, p = 0.746, partial $\eta^2$ = 0.001) and at the end of the isolation period (F = 0.361, p = 0.55, partial $\eta^2$ = 0.004). However, at the end of 3 months, the simple effect of the group was significant (F = 5.765, p = 0.018, partial $\eta^2$ = 0.056), which eventually became nonsignificant at the end of 6 months (F = 3.659, p = 0.059, partial $\eta^2$ = 0.036). The simple effect of time was significant in the intervention (F = 59.184, p < 0.001, partial $\eta^2$ = 0.651) and control (F = 29.001, p < 0.001, partial $\eta^2$ = 0.478) groups. Post-hoc comparisons showed that HbA1c levels in the intervention group were lower at the end of 3 and 6 months than at baseline and at the end of isolation (p < 0.05), whereas those in the control group at the end of 6 months were lower than that at the end of 3 months and lower than those at the baseline and end of isolation (p < 0.05; Table 3).

## Discussion

The COVID-19 pandemic has been an unprecedented global health concern affecting millions worldwide. Patients with diabetes are more susceptible to COVID-19 and reportedly have a

**Table 2. The follow-up data of the two groups.**

| Characteristics | 21 days | | | 3 months | | | 6 months | | |
|---|---|---|---|---|---|---|---|---|---|
| | Control | Intervention | P vs. control | Control | Intervention | P-value | Control | Intervention | P-value |
| HbA1c(%) | 8.41 (1.21) | 8.57 (1.43) | 0.55 | 7.25 (1.78)[b] | 6.60 (1.31)[bc] | 0.04 | 6.66 (1.63)[b] | 6.14 (1.05)[b] | 0.13 |
| Change vs. baseline | −0.09 (1.52) | 0.01 (1.88) | | −1.17 (1.71) | −1.95 (1.71) | | −1.84 (1.55) | −2.41 (1.38) | |
| FBG (mmol/L) | 8.86 (7.77–9.77) | 6.52 (5.53–8.17)[bd] | <0.01 | 6.40 (5.35–8.50)[b] | 5.45 (4.81–7.01)[bc] | 0.02 | 5.99 (4.68–7.67)[b] | 5.58 (4.88–6.78)[b] | 0.65 |
| Change vs. baseline | −0.17 (1.66) | −1.88 (1.36) | | −2.08 (2.15) | −2.57 (2.24) | | −2.83 (2.03) | −2.74 (1.96) | |
| PBG (mmol/L) | 12.37 (11.52–13.18) | 12.07 (11.05–13.80) | 0.60 | 7.88 (6.97–9.06)[b] | 7.74 (6.91–8.62)[b] | 0.37 | 7.58 (6.79–9.19)[b] | 6.78 (6.22–7.70)[bd] | <0.01 |
| Change vs. baseline | −0.53 (1.86) | −0.56 (2.10) | | −4.25 (2.01) | −4.88 (1.79) | | −4.86 (2.10) | −5.52 (2.14) | |
| BMI (kg/m$^2$) | 28.73 (2.97) | 28.72 (2.61) | 0.93 | 28.82 (2.57) | 27.10 (2.61)[bd] | <0.01 | 27.36 (1.90)[b] | 25.49 (2.35)[bd] | <0.01 |
| Change vs. baseline | −0.32 (4.81) | −0.54 (3.85) | | −0.60 (4.41) | −2.16 (3.65) | | −1.68 (3.95) | −3.77 (3.38) | |
| Waist-to-hip ratio | 0.95 (0.04) | 0.94 (0.04) | 0.82 | 0.95 (0.04) | 0.96 (0.05) | 0.77 | 0.91 (0.09)[a] | 0.86 (0.11)[bc] | 0.03 |
| Change vs. baseline | 0.003 (0.06) | −0.02 (0.06) | | 0.01 (0.07) | −0.001 (0.06) | | −0.04 (0.09) | −0.09 (0.10) | |
| TG (mmol/L) | 2.10 (1.83–2.40) | 1.96 (1.63–2.17) | 0.10 | 1.65 (1.52–2.20)[a] | 1.63 (1.45–1.99)[a] | 0.40 | 1.67 (1.51–1.98)[b] | 1.56 (1.43–1.83)[bc] | 0.02 |
| Change vs. baseline | 0.01 (0.57) | −0.11 (0.55) | | −0.20 (0.65) | −0.25 (0.67) | | −0.26 (0.54) | −0.39 (0.61) | |
| LDL-C (mmol/L) | 3.65 (0.33) | 3.63 (0.34) | 0.78 | 3.47 (0.41)[a] | 3.49 (0.37)[b] | 0.70 | 3.39 (0.50)[b] | 3.03 (0.58)[bd] | <0.01 |
| Change vs. baseline | 0.05 (0.39) | −0.07 (0.41) | | −0.13 (0.41) | −0.21 (0.45) | | −0.21 (0.50) | −0.68 (0.66) | |
| SDS | 44.02 (9.71)[a] | 41.19 (9.38) | 0.10 | 39.02 (10.12) | 40.08 (10.64) | 0.48 | 39.94 (9.50) | 37.08 (9.16)[a] | 0.09 |
| Change vs. baseline | 3.85 (12.19) | 0.56 (12.49) | | −1.15 (12.95) | −0.56 (12.70) | | −0.23 (11.24) | −3.56 (12.41) | |
| Blood pressure systolic (mm Hg) | 134.70 (122.05–144.25) | 136.75 (125.50–144.43) | 0.48 | 130.50 (124.70–141.90) | 131.20 (122.70–142.55) | 0.97 | 137.70 (126.45–143.65) | 132.55 (125.68–141.98) | 0.99 |
| Change vs. baseline | −2.71 (16.44) | −0.94 (15.89) | | −3.43 (15.50) | −3.34 (16.21) | | −0.56 (14.67) | −2.47 (14.53) | |
| Blood pressure, diastolic (mm Hg) | 86.80 (80.80–91.95) | 87.75 (80.63–93.33) | 0.61 | 85.30 (80.40–89.25) | 85.90 (82.98–91.68) | 0.54 | 86.50 (82.20–90.85) | 85.50 (81.05–91.45) | 0.64 |
| Change vs. baseline | −0.20 (9.15) | 1.20 (10.02) | | −1.76 (8.98) | 1.13 (9.06) | | −0.52 (8.75) | −0.11 (9.73) | |
| TC (mmol/L) | 4.89 (4.51–5.44) | 4.85 (4.62–5.26) | 0.46 | 5.15 (4.64–5.42) | 5.00 (4.46–5.49) | 0.34 | 4.96 (4.52–5.42) | 4.84 (4.57–5.51) | 0.78 |
| Change vs. baseline | 0.0002 (0.66) | −0.06 (0.71) | | 0.11 (0.71) | 0.02 (0.72) | | −0.01 (0.74) | 0.02 (0.72) | |
| HDL-C (mmol/L) | 1.23 (1.10–1.34) | 1.26 (1.12–1.39) | 0.64 | 1.32 (1.14–1.41) | 1.26 (1.10–1.36) | 0.27 | 1.20 (1.07–1.37) | 1.22 (1.12–1.38) | 0.53 |
| Change vs. baseline | −0.04 (2.23) | 0.03 (0.21) | | 0.0002 (0.22) | 0.03 (0.22) | | −0.05 (0.25) | 0.04 (0.23) | |
| BUN (mmol/L) | 5.70 (5.05–6.40) | 6.00 (5.35–6.60) | 0.21 | 5.70 (5.40–6.20) | 5.80 (5.40–6.53) | 0.43 | 5.90 (5.20–6.10) | 6.05 (5.18–6.50) | 0.24 |
| Change vs. baseline | −0.17 (0.87) | 0.02 (0.87) | | −0.12 (0.81) | −0.01 (0.83) | | −0.16 (0.93) | −0.01 (0.84) | |
| Cr (mmol/L) | 68.70 (62.7–73.8) | 67.20 (61.80–74.10) | 0.82 | 65.80 (57.45–73.40) | 67.15 (60.83–73.40) | 0.67 | 61.50 (56.65–66.55) | 67.85 (58.95–6.50) | 0.76 |
| Change vs. baseline | 2.57 (9.55) | 2.32 (10.16) | | 0.34 (10.86) | 1.83 (11.56) | | −2.80 (11.19) | 1.77 (10.62) | |
| e-GFR(ml/min) | 91.95 (9.65) | 91.30 (9.46) | 0.70 | 89.84 (8.75) | 89.93 (6.94) | 0.95 | 90.39 (8.36) | 89.97 (9.59) | 0.82 |
| Change vs. baseline | 1.88 (13.82) | −0.23 (14.84) | | −0.23 (12.98) | −1.59 (13.21) | | 0.32 (13.93) | −1.56 (13.11) | |

Data are presented as the means (SD) for the normally distributed variables, the median (IQR) for the non-normally distributed variables or the number of participants (%). FBG: fasting blood glucose, PBG: postprandial blood glucose, HbA1c: glycated hemoglobin, BMI: Body Mass Index, TC: total cholesterol, TG: triglyceride, LDL-C: low-density lipoprotein cholesterol, HDL-C: high-density lipoprotein cholesterol, BUN: blood urea nitrogen, Cr: creatinine, e-GFR: estimated glomerular filtration rate, SDS: Self-Rating Depression Scale. In comparison with baseline, "a" indicates $p < 0.05$ versus baseline; "b" indicates $p < 0.01$ versus baseline; "*" indicates $p < 0.05$ versus control group; and "**" indicates $p < 0.05$ versus control group.

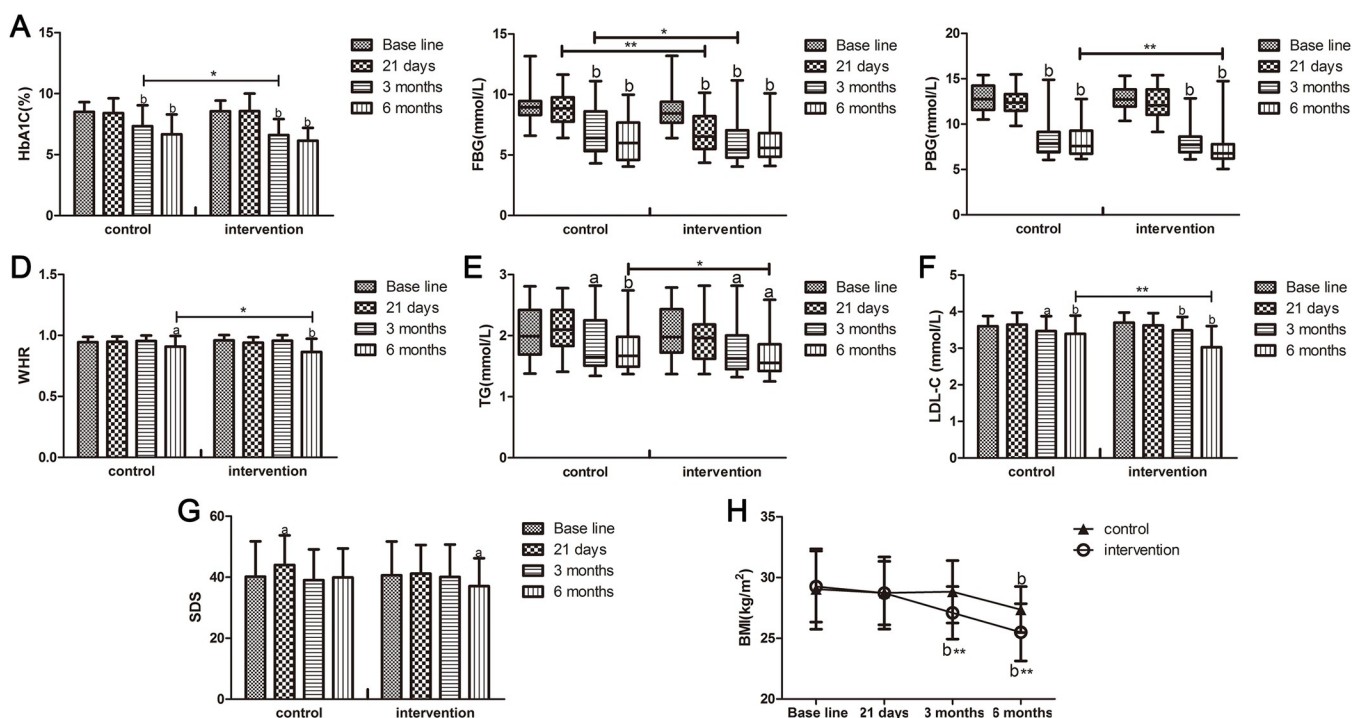

**Fig 2. Targets of the study.** (A-H): Changes in the HbA1c, FBG, PBG, TG, and LDL-C levels as well as SDS, WHR, and BMI over 6 months of the study in the control and intervention groups. HbA1c: glycated hemoglobin, FBG: fasting blood glucose, PBG: postprandial blood glucose, WHR: waist-to-hip ratio, TG: triglyceride, LDL-C: low-density lipoprotein cholesterol, SDS: Self-Rating Depression Scale, BMI: body mass index. "a" indicates p < 0.05 vs. baseline; "b" indicates p < 0.01 vs. baseline; "*" indicates p < 0.05 vs. control group; "**" indicates p < 0.05 vs. control group.

rapid disease progression, high severity, and high fatality rate after infection [17]. The upregulation of the angiotensin-converting enzyme (ACE)-2 receptor gene in the cardiomyocytes of patients with diabetes mellitus along with nonenzymatic glycation may increase the susceptibility to COVID-19 infection in patients with diabetes by promoting the entry of severe acute respiratory syndrome coronavirus 2 into the cell [18]. Furthermore, the immune dysfunction, proinflammatory cytokine environment, hypoglycemic state, and coagulopathy of patients with diabetes contribute to a poor prognosis and complications of COVID-19 by increasing the risk of mechanical ventilation, shock, and multiple organ failure, eventually leading to death [19–23]. Sardu showed that early glycemic control reduces adverse events in hospitalized patients with hyperglycemic COVID-19 with or without a previous diagnosis of diabetes [23]. Therefore, ensuring competent management of diabetes and taking scientific and effective measures during the pandemic is crucial to improving immunity and reducing the risk of infection in patients with diabetes.

**Table 3. A comparison of the primary outcome, HbA1c, throughout the 6-month follow-up.**

|  | n | Baseline | 21 days | 3 months | 6 months | $F$ | $P$ | Partial $\eta^2$ |
|---|---|---|---|---|---|---|---|---|
| Intervention | 52 | 8.56 ± 0.88 | 8.57 ± 1.43 | 6.60 ± 1.31[ab] | 6.14 ± 1.05[ab] |  |  |  |
| Control | 47 | 8.50 ± 0.80 | 8.41 ± 1.21 | 7.34 ± 1.71[ab] | 6.66 ± 1.63[abc] |  |  |  |
| Group main effect |  |  |  |  |  | 2.558 | 0.113 | 0.026 |
| Time main effect |  |  |  |  |  | 86.089 | <0.001 | 0.470 |
| Group*Time interaction effect |  |  |  |  |  | 3.498 | 0.019 | 0.035 |

Compared with baseline, "a" indicates p < 0.05; compared with 21 days, "b" indicates p < 0.05; compared with the end of 3 months, "c" indicates p < 0.05.

However, the stringent pandemic control measures have posed new challenges to the management and follow-up of patients with chronic illnesses. In this regard, telemedicine offers patients an opportunity for remote diagnosis, treatment, and consultation; reduces medical expenses; and prevents cross-infection during outpatient visits. Few studies have indicated that telemedicine should be actively used to control the development of chronic illnesses during an epidemic [24]. Moreover, Bahl et al. suggested that telemedicine should be explicitly promoted as an alternative to traditional medical care during the COVID-19 pandemic, particularly during isolation [25]. Currently, several studies on telemedicine management of diabetes have reported favorable outcomes; however, these studies mostly comprised older patients. There has been a recent increase in the incidence of T2DM in younger patients who belong to the age group that was considerably affected by the lockdown measures. Therefore, we assessed the efficacy of telemedicine in obese and overweight young and middle-aged patients with diabetes. Although obesity is not the only risk factor for T2DM, it can significantly increase the risk of complications in patients with T2DM [26]. Therefore, we focused on the application of telemedicine in the glycemic control of obese and overweight patients with T2DM during and after a 21-day isolation period.

In our study, the intervention and control groups managed blood glucose levels through telemedicine and outpatient follow-up, respectively. At the end of the isolation period, FBG level in the intervention group was lower than that in the control group and the baseline value. After 3 months, both groups showed significant reductions in HbA1c and FBG levels, but the reduction was greater in the intervention group. Likewise, at the end of 6 months, PBG level decreased more significantly in the intervention group than in the control group. Additionally, 10 and 6 patients in the control and intervention groups, respectively, required adjustment of drug dosage for titration to achieve glycemic control during the study. At the end of 6 months, the rate of discontinuation of antidiabetic medications was 11.5% in the intervention group versus 2.1% in the control group. None of the patients in either group experienced any serious adverse events or aggravation of complications. Further, the intervention group showed significantly fewer hypoglycemic events than the control group (n = 8 vs. n = 13). This difference may be attributed to the fact that the medical team could identify the risk of hypoglycemia at an appropriate time in the intervention group through telemedicine and take appropriate measures. These results revealed the positive effect of telemedicine in ensuring adequate glycemic control in patients with T2DM and suggested that the use of telemedicine should be promoted and popularized, especially when diligent physical monitoring is not possible.

Obesity and T2DM are closely associated and are growing global public health problems. Weight gain is an independent risk factor for T2DM, and 87.5% of adults with diabetes are overweight or obese [27]. The recent COVID-19 isolation measures have further reduced physical activity and promoted sedentary behavior. Renzo et al. studied the eating habits of and lifestyle changes in 3,533 respondents in Italy during the COVID-19 pandemic and reported that weight gain was observed in 48.6% of the study population [28]. These results highlight the fact that weight control is a challenging task and is particularly important in managing diabetes during the COVID-19 epidemic. In our study, at the end of 3 and 6 months, BMI was significantly lower in the intervention group than in the control group; similarly, at the end of 6 months, WHR reduced significantly in the intervention group compared with the control group. This indicates that diabetes management through telemedicine can effectively help in weight control and abdominal obesity improvement. Both groups received diet and exercise instructions; however, the medical team, including doctors, nurses, and nutritionists, could monitor the diet of patients in the intervention group more frequently through telemedicine, allowing for more precise guidance. Strategies such as weighing frequently, recording energy intake, and tracking physical activity are currently used to improve compliance;

however, these strategies are inadequate to achieve significant weight loss without feedback [29]. Weight and blood glucose management can be more effective by obtaining timely feedback from patients. In this study, we provided individualized, simple, and feasible home exercise guidance according to the actual energy intake of different patients. The majority of patients (45/52) could complete most of the prescribed exercise at the end of the study and showed significant weight loss. In addition, the fast food consumed by young and middle-aged people during their daily work is often calorie-rich junk food. During the 21-day isolation and under expert dietary guidance, the intervention group could improve their dietary structure, which may also be one of the reasons for the efficient glycemic and weight control in these patients.

Depression is an independent risk factor for T2DM [30]. Depression decreases dietary and treatment adherence in patients with diabetes, thereby affecting glycemic control, aggravating symptoms, and reducing their quality of life [31, 32]. According to a meta-analysis of 11 longitudinal studies, depression can considerably increase the risk of developing T2DM by 37% [33]. Further, several studies have shown that the incidence of depression in patients with diabetes is significantly higher than that in normal individuals. Khaledi et al. reported that approximately 25% of adults with T2DM suffer from depression [34]. Zhong collected the demographic and lifestyle data of 19,802 participants from 34 provinces in China during the COVID-19 pandemic through a web-based survey and revealed that participants with chronic illnesses had a 93% increased risk of high anxiety levels [35]. Likewise, Xiang et al. highlighted the need for urgent interventions for the mental health of patients with chronic illnesses during the COVID-19 pandemic [36]. In the current study, we assessed the effect of telemedicine in addition to glycemic and weight control on diabetes-related anxiety symptoms in obese and overweight young and middle-aged patients with T2DM. We observed that after the 21-day isolation period, the SDS score of the control group was significantly higher than the baseline value ($p < 0.05$). After the 6-month follow-up, the SDS score of the intervention group significantly decreased compared with the baseline value ($p < 0.05$) and was lower than that of the control group, although the difference was not statistically significant. Based on these results, we can infer that depressive symptoms in patients with chronic illnesses could be relieved to some extent via psychological counseling and communication through telemedicine. Additionally, the favorable results for telemedicine could be attributed to the fact that the study population included young and middle-aged patients who have higher acceptance of modern communication devices such as smartphones than other age groups.

Recently, the role of multifaceted management of patients with T2DM has been repeatedly emphasized [37]. The telemedicine measures provided in this study focused more on the popularization of diabetes mellitus-related knowledge, diet/exercise guidance, blood glucose monitoring, and adjustment of glucose-lowering regimen. We believe that patients should receive more comprehensive multifactorial control, if conditions permit. However, further research is warranted in this regard.

This study has some limitations. First, it is a single-center study. Second, the study population was limited to individuals who could use mobile phones and internet independently, which may have affected the applicability of our results to patients with T2DM who require constant monitoring and cannot access modern technology. Third, the authors must acknowledge the shortcomings of the missing data in this study, in which 120 patients were enrolled and only 99 completed the 6-month follow-up. Due to these missing data (17.5% of the study population), the findings of this study should be considered with caution.

In conclusion, this study revealed that telemedicine had a positive effect on glycemic control and weight management in obese and overweight young and middle-aged patients with T2DM during the COVID-19 pandemic. Moreover, follow-up through telemedicine was

superior to that through conventional outpatient clinic appointment in controlling depressive symptoms during the pandemic. These results are extremely encouraging and corroborate the importance of telehealth measures to provide more professional, safe, economic, and personalized management and improve the ability of patients to self-manage their conditions, resulting in a higher quality of life. Furthermore, this study provides a basis for the development of an individualized telemedicine management model for diabetes.

## Supporting information

**S1 Checklist. CONSORT 2010 checklist of information to include when reporting a randomised trial**∗**.**
(DOC)

**S1 Protocol.**
(PDF)

## Acknowledgments

We thank the participants who took part in the randomized controlled trial.

## Author Contributions

**Conceptualization:** Wenwen Yin.

**Data curation:** Wenwen Yin, Yawen Liu, Hao Hu.

**Formal analysis:** Wenwen Yin, Yawen Liu.

**Investigation:** Wenwen Yin, Hao Hu, Yuanyuan Liu.

**Methodology:** Wenwen Yin, Jin Sun.

**Project administration:** Yawen Liu.

**Supervision:** Yawen Liu, Zhaoling Wang.

**Writing – original draft:** Wenwen Yin.

**Writing – review & editing:** Zhaoling Wang.

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
