## [Decision Letter · Decision Letter 0]

5 Jul 2022

PONE-D-22-12805Telemedicine management of type 2 diabetes mellitus in obese and overweight young and middle-aged patients during COVID-19 outbreak: A single-center, prospective, randomized control studyPLOS ONE

Dear Dr. Wang,

Thank you for submitting your manuscript to PLOS ONE. After careful consideration, we feel that it has merit but does not fully meet PLOS ONE’s publication criteria as it currently stands. Therefore, we invite you to submit a revised version of the manuscript that addresses the points raised during the review process. I advice to submit a revised version of your manuscript, after have addressed all issues raised by reviewers. 

We look forward to receiving your revised manuscript.

Kind regards,

Ferdinando Carlo Sasso, PhD, MD

Academic Editor

PLOS ONE

Journal Requirements:

Additional Editor Comments:

The manuscript was reviewed by external reviewers. Several issues have been raised, but if the authors can address them, I suggest submitting a revised version based on peer reviewers' comments.

Reviewers' comments:

Reviewer's Responses to Questions

**Comments to the Author**

1. Is the manuscript technically sound, and do the data support the conclusions?

Reviewer #1: Yes

Reviewer #2: Partly

Reviewer #3: Partly

2. Has the statistical analysis been performed appropriately and rigorously? 

Reviewer #1: Yes

Reviewer #2: No

Reviewer #3: No

3. Have the authors made all data underlying the findings in their manuscript fully available?

Reviewer #1: Yes

Reviewer #2: Yes

Reviewer #3: No

4. Is the manuscript presented in an intelligible fashion and written in standard English?

Reviewer #1: Yes

Reviewer #2: No

Reviewer #3: No

5. Review Comments to the Author

Reviewer #1: I read with great interest the paper “Telemedicine management of type 2 diabetes mellitus in obese and overweight young and middle-aged patients during COVID-19 outbreak: A single-center, prospective, randomized control study" by Yin et al.

The article is well written. The paper has a good design. The article is logically divided into sections and subsections. Telemedicine as a screening tool, in particularly during covid pandemic, has been much debated and applied and a lot of papers have already been published.

Comments:

1. Line 81-82: “to help the patients to achieve blood glucose control”. I understand that the paper mainly focusses on glycaemic control, however this sentence is reductive, since, in diabetic patients, the real benefit is provided by the multifactorial control rather than one (doi: 10.1186/s12933-021-01343-1). Moreover, telemedicine as a screening tool has proven its efficacy both during covid pandemic and before, not representing a novel option (doi: 10.1155/2020/9036847; doi: 10.1002/dmrr.3113).

2. As this is a clinical trial, missing patients should be less than 5%. In this specific case 120 patients were enrolled and only 99 finished the period of follow up (missing 17.5%). The problem of missing data is of particular importance due to it introducing bias and leading to a loss of power, inefficiencies, and false positive findings (Type I Error). It MUST be reported in the limit of the study. Moreover, it should be reported in the appropriate section what happened to these patients (retrieved consent, etc…).

3. Discussion: it has been pointed out that depression may affect diabetes. Another important element is the environment. In fact, family dietary habit, patients dietary habit (the patients where isolated at home for 21 days, but most of them may be consumers of junk food during daytime due to work) etc… should be also taken into consideration.

Reviewer #2: Major concerns:

It is critical to calculate the sample size when you design the study, especially for the confirmatory analysis of the effects. This is a major flaw of the study.

The statistical analysis is under development. For the repeated measures, better to use longitudinal models.

T-test was mentioned but was not used. Line 173, it should be Wilcoxon signed rank test.

P values need to be adjusted for multiple tests. How many tests will lose significance after adjustment? The conclusions should be re-evaluated after correct statistics are performed.

The writing needs to be polished by a native speaker. I just list a few in the abstract here. The whole manuscript needs to be rewritten.

Line 36 the “effects” of Tele

Line 37 on diabetes management for patients with obesity/overweight and T2DM

Line 39 with T2DM “were” enrolled

Line 42 all enrolled participants

Line 46 there are 99 participants … (52 participants in the Tele group, and 47 participants in the control group). Better not start a sentence with a number.

Line 51 “decreased significantly in the intervention group than in the control group” is not clear for comparative. Better rewrite.

Line 52 “decline degree”? You mean reduction?

Line 53 “the reduction of” LDL…

Line 55 decreased “stronger”?

“than the control group”, “than that of the control group”, “than that in the control group” …

Report p value in exact numbers. Do not use p<0.05 or p<0.01.

Report statistics. How much decrease?

Minor concerns:

Table 1 & 2 add p values for comparisons

It is better to show reduction from baseline in table 2.

Page 7. Add space between medians and brackets, and space after coma.

Figure 1 how are the participants stratified? “Analyzed date” is not correct.

Figure 2 It is not clear whether the bars are means or medians. How about the error bars? Better be consistent with Table 2. If you decide to use median, then present medians in the table and in the figure. Do not mix usage of parametric and nonparametric methods.

Reviewer #3: Dear editor,

About the manuscript entitled “Telemedicine management of type 2 diabetes mellitus in obese and overweight young and middle-aged patients during COVID-19 outbreak: A single-center, prospective, randomized control study”, enclosed my suggestions:

INTRODUCTION:

In the text, less has been reported about the cellular mechanisms of viral entrance, replication and pathogenesis, and more less data about the clinical outcomes of COVID-19 in patients at higher risk as reported for patients with hypertension and type 2 diabetes mellitus. In this setting, I would remember:

-in the pathogenesis of SARS/COV2 infection, the role played by serin proteasis expression (TMPRSS2) in humans’ cells as main cause of the entrance and replication of SARS/COV2 (miR-98 Regulates TMPRSS2 Expression in Human Endothelial Cells: Key Implications for COVID-19. Biomedicines. 2020 Oct 30;8(11):462. doi: 10.3390/biomedicines8110462), and parallel the different cellular expression of ACE2 in humans, and its negative effects on clinical outcomes in humans (Cardiovasc Diabetol. 2021 May 7;20(1):99. doi: 10.1186/s12933-021-01286-7). Please describe this point and refer to the suggested reference.

-please introduce the population at higher risk of worse prognosis as the hypertensive patients (Could anti-hypertensive drug therapy affect the clinical prognosis of hypertensive patients with COVID-19 infection? Data from centers of southern Italy. J Am Heart Assoc. 2020 Jul 7:e016948. doi: 10.1161/JAHA.120.016948), and the diabetics (Outcomes in Patients With Hyperglycemia Affected by COVID-19: Can We Do More on Glycemic Control? Diabetes Care. 2020 Jul;43(7):1408-1415. doi: 10.2337/dc20-0723; Impact of diabetes mellitus on clinical outcomes in patients affected by Covid-19. Cardiovasc Diabetol. 2020 Jun 11;19(1):76. doi: 10.1186/s12933-020-01047-y). Indeed, these patients are those at higher risk for COVID19, ICU admission and deaths. Please explain this point in the text, and the association between these cohorts of patients and the worse prognosis. What is your opinion? Please discuss it.

METHODS:

-How did you diagnose T2DM? what were the cutoff Hb1Ac values for inclusion/exclusion study criteria?

-report the full data about the industry implied/used for laboratory analysis.

-report a full descriptive sub-chapter about the laboratory diagnosis of COVID-19 infection. Report in detail study population, inclusion vs. exclusion criteria.

-How did you diagnose and monitor study endpoints? Please discuss it, including all techniques and methods for measuring the study outcomes.

RESULTS:

You wrote that “Data is presented as the median (Quartile 1-Quartile 3) or the number of participants (%)”. It does not look. It looks as the data is presented as mean ± standard deviations and/or as median values. Please correct and discuss it.

Please include all the measures units of study variables. The same in tables.

The study cohorts are represented by a low number of subjects. In my opinion the statistics could be under powered, and this limiting the study results. How did you calculate the sample size? Please address the question.

Tables and figures are of poor quality. Improve it.

The legend of figure 2 is not clear. What does it mean “a P﹤0.05 ,b P<0.01; between–group comparisons, * P<0.05, ** P<0.01”. I do not understand. Could it be: * p<0.05 at baseline; ** p<0.05 at follow-up end? Please respond and report the right correction.

Please add the column with p values by comparing cohorts of study.

Add the full medical therapy as anti-SARS-CoV-2, and anti-diabetic medications. Could it affect clinical study outcomes? Please discuss it.

how many patients were under insulin therapy in the cohorts during hospitalizations?

what was the rate of anti-diabetic medications discontinuation during hospitalization? What was the mean glycemia during hospital discharge?

what were the patients under anti-IL6 (tolicizumab) therapy? Indeed, also in treated patients there could be a loss of effect caused by the negative factors as hyperglycemia (Negative impact of hyperglycaemia on tocilizumab therapy in Covid-19 patients. Diabetes and Metabolism 2020; doi: 10.1016/j.diabet.2020.05.005). please discuss this point and the suggested reference.

Indeed, the hyperglycemia is a recognized and reported factor of worse prognosis in the patients with COVID-19 (Hyperglycaemia on admission to hospital and COVID-19. Diabetologia. 2020 Jul 6:1-2. doi: 10.1007/s00125-020-05216-2). Please discuss and clarify this point in the text.

Finally, there are not data about the vaccinated patients under hyperglycemia and the COVID-19. Notably, again the hyperglycemia could reduce the efficacy of vaccination (The CAVEAT study. Diabetes Obes Metab. 2022 Jan;24(1):160-165. doi: 10.1111/dom.14547; Nat Commun. 2022 Apr 28;13(1):2318. doi: 10.1038/s41467-022-30068-2). Please discuss it.

DISCUSSION:

It is too long, and not well focused on main study outcomes. Please rewrite it.

increase quality of tables and figures.

Improve English form of the text.

6. PLOS authors have the option to publish the peer review history of their article (what does this mean?). If published, this will include your full peer review and any attached files.

Reviewer #1: No

Reviewer #2: No

Reviewer #3: No

---

## [Author Response · Author response to Decision Letter 0]

19 Aug 2022

We thank reviewers and editor for valuable comments, and we've responded to reviewers and editor comments in document "Response to Reviewers".

---

## [Decision Letter · Decision Letter 1]

22 Aug 2022

PONE-D-22-12805R1Telemedicine management of type 2 diabetes mellitus in obese and overweight young and middle-aged patients during COVID-19 outbreak: A single-center, prospective, randomized control studyPLOS ONE

Dear Dr. Wang,

Thank you for submitting your manuscript to PLOS ONE. After careful consideration, we feel that it has merit but does not fully meet PLOS ONE’s publication criteria as it currently stands. Therefore, we invite you to submit a revised version of the manuscript that addresses the points raised during the review process.

ACADEMIC EDITOR:

We look forward to receiving your revised manuscript.

Kind regards,

Ferdinando Carlo Sasso, PhD, MD

Academic Editor

PLOS ONE

Journal Requirements:

Additional Editor Comments:

Both reviewers stated that all issues raised were addressed by authors.

However, Reviewer 3 underlines that the manuscript needs revision by a native English speaker.

Reviewers' comments:

Reviewer's Responses to Questions

**Comments to the Author**

1. If the authors have adequately addressed your comments raised in a previous round of review and you feel that this manuscript is now acceptable for publication, you may indicate that here to bypass the “Comments to the Author” section, enter your conflict of interest statement in the “Confidential to Editor” section, and submit your "Accept" recommendation.

Reviewer #1: All comments have been addressed

Reviewer #3: All comments have been addressed

2. Is the manuscript technically sound, and do the data support the conclusions?

Reviewer #1: Yes

Reviewer #3: Yes

3. Has the statistical analysis been performed appropriately and rigorously? 

Reviewer #1: Yes

Reviewer #3: Yes

4. Have the authors made all data underlying the findings in their manuscript fully available?

Reviewer #1: Yes

Reviewer #3: No

5. Is the manuscript presented in an intelligible fashion and written in standard English?

Reviewer #1: Yes

Reviewer #3: No

6. Review Comments to the Author

Reviewer #1: The author managed to answer to all the issue I raised. The paper has much improved and can now be further processed for publication

Reviewer #3: The authors responded to reviewers comments.

Finally I ask to improve the english form of the text.

Please submit the revised form for a possible publication

in the journal .

7. PLOS authors have the option to publish the peer review history of their article (what does this mean?). If published, this will include your full peer review and any attached files.

Reviewer #1: No

Reviewer #3: No

---

## [Author Response · Author response to Decision Letter 1]

29 Aug 2022

Thank you for helping us a lot, your suggestion is of great help to us. We again invited a native English speaker to revise the language of the article.

---

## [Decision Letter · Decision Letter 2]

7 Sep 2022

PONE-D-22-12805R2Telemedicine management of type 2 diabetes mellitus in obese and overweight young and middle-aged patients during COVID-19 outbreak: a single-center, prospective, randomized control studyPLOS ONE

Dear Dr. Wang,

Thank you for submitting your manuscript to PLOS ONE. After careful consideration, we feel that it has merit but does not fully meet PLOS ONE’s publication criteria as it currently stands. Therefore, we invite you to submit a revised version of the manuscript that addresses the points raised during the review process.

We look forward to receiving your revised manuscript.

Kind regards,

Ferdinando Carlo Sasso, PhD, MD

Academic Editor

PLOS ONE

Journal Requirements:

Additional Editor Comments:

The authors have to addressed the issues raised by the statistician.

Reviewers' comments:

Reviewer's Responses to Questions

**Comments to the Author**

1. If the authors have adequately addressed your comments raised in a previous round of review and you feel that this manuscript is now acceptable for publication, you may indicate that here to bypass the “Comments to the Author” section, enter your conflict of interest statement in the “Confidential to Editor” section, and submit your "Accept" recommendation.

Reviewer #1: All comments have been addressed

Reviewer #3: All comments have been addressed

2. Is the manuscript technically sound, and do the data support the conclusions?

Reviewer #1: Yes

Reviewer #3: Yes

3. Has the statistical analysis been performed appropriately and rigorously? 

Reviewer #1: Yes

Reviewer #3: Yes

4. Have the authors made all data underlying the findings in their manuscript fully available?

Reviewer #1: Yes

Reviewer #3: Yes

5. Is the manuscript presented in an intelligible fashion and written in standard English?

Reviewer #1: Yes

Reviewer #3: Yes

6. Review Comments to the Author

Some comments are still not addressed. 

1. Add p values to table 1. 

2. For all tables, add space between mean /median and the open parentheses. Add space after comma.

7. PLOS authors have the option to publish the peer review history of their article (what does this mean?). If published, this will include your full peer review and any attached files.

Reviewer #1: No

Reviewer #3: No

---

## [Author Response · Author response to Decision Letter 2]

9 Sep 2022

1. Add p values to table 1.

Response: We have added p values to Table 1.

2. For all tables, add space between mean /median and the open parentheses. Add space after comma.

Response: Thank you very much for the reminder! In Tabel 1 and Table 2, we have added space between mean /median and the open parentheses, and we have changed the expression of the median (IQR) to be consistent with that in "Result".

---

## [Decision Letter · Decision Letter 3]

13 Sep 2022

Telemedicine management of type 2 diabetes mellitus in obese and overweight young and middle-aged patients during COVID-19 outbreak: a single-center, prospective, randomized control study

PONE-D-22-12805R3

Dear Dr. Wang,

We’re pleased to inform you that your manuscript has been judged scientifically suitable for publication and will be formally accepted for publication once it meets all outstanding technical requirements.

Kind regards,

Ferdinando Carlo Sasso, PhD, MD

Academic Editor

PLOS ONE

Additional Editor Comments (optional):

The authors addressed the issues raised by all reviewers. The revised paper can be accepted for publication.

Reviewers' comments:

Reviewer's Responses to Questions

**Comments to the Author**

1. If the authors have adequately addressed your comments raised in a previous round of review and you feel that this manuscript is now acceptable for publication, you may indicate that here to bypass the “Comments to the Author” section, enter your conflict of interest statement in the “Confidential to Editor” section, and submit your "Accept" recommendation.

Reviewer #1: All comments have been addressed

Reviewer #2: All comments have been addressed

Reviewer #3: (No Response)

2. Is the manuscript technically sound, and do the data support the conclusions?

Reviewer #1: Yes

Reviewer #2: (No Response)

Reviewer #3: (No Response)

3. Has the statistical analysis been performed appropriately and rigorously? 

Reviewer #1: Yes

Reviewer #2: (No Response)

Reviewer #3: (No Response)

4. Have the authors made all data underlying the findings in their manuscript fully available?

Reviewer #1: Yes

Reviewer #2: (No Response)

Reviewer #3: (No Response)

5. Is the manuscript presented in an intelligible fashion and written in standard English?

Reviewer #1: Yes

Reviewer #2: (No Response)

Reviewer #3: (No Response)

6. Review Comments to the Author

Reviewer #1: The authors managed to respond to all my queries. The paper has much improved and can be further processed for publication

Reviewer #2: (No Response)

Reviewer #3: The authors responded to all’ comments.

In my opinion we could accept the article for a possible

Publication in the journal.

7. PLOS authors have the option to publish the peer review history of their article (what does this mean?). If published, this will include your full peer review and any attached files.

Reviewer #1: No

Reviewer #2: No

Reviewer #3: No

---

## [Editor Report · Acceptance letter]

19 Sep 2022

PONE-D-22-12805R3 

Telemedicine management of type 2 diabetes mellitus in obese and overweight young and middle-aged patients during COVID-19 outbreak: a single-center, prospective, randomized control study 

Dear Dr. Wang:

I'm pleased to inform you that your manuscript has been deemed suitable for publication in PLOS ONE. Congratulations! Your manuscript is now with our production department. 

Kind regards, 

on behalf of

Professor Ferdinando Carlo Sasso 

Academic Editor

PLOS ONE